# On-CMOS Image Sensor Processing for Lane Detection

**DOI:** 10.3390/s21113713

**Published:** 2021-05-26

**Authors:** Soyeon Lee, Bohyeok Jeong, Keunyeol Park, Minkyu Song, Soo Youn Kim

**Affiliations:** Department of Semiconductor Science, Dongguk University, Seoul 04620, Korea; soyeon1104@dgu.ac.kr (S.L.); jbo20018@dongguk.edu (B.J.); kj170494@dongguk.edu (K.P.); mksong@dongguk.edu (M.S.)

**Keywords:** CMOS image sensor, computer vision, edge detection, low power consumption, single-slope ADC

## Abstract

This paper presents a CMOS image sensor (CIS) with built-in lane detection computing circuits for automotive applications. We propose on-CIS processing with an edge detection mask used in the readout circuit of the conventional CIS structure for high-speed lane detection. Furthermore, the edge detection mask can detect the edges of slanting lanes to improve accuracy. A prototype of the proposed CIS was fabricated using a 110 nm CIS process. It has an image resolution of 160 (H) × 120 (V) and a frame rate of 113, and it occupies an area of 5900 μm × 5240 μm. A comparison of its lane detection accuracy with that of existing edge detection algorithms shows that it achieves an acceptable accuracy. Moreover, the total power consumption of the proposed CIS is 9.7 mW at pixel, analog, and digital supply voltages of 3.3, 3.3, and 1.5 V, respectively.

## 1. Introduction

The core technology of self-driving cars is a vision sensor equipped with a lane departure warning system (LDWS), which has attracted much attention [1,2,3,4]. In the LDWS, a camera installed in the vehicle acts as a vision sensor that can detect a lane and notify a driver when there is a risk of unintended lane departure. Figure 1a shows a system flow chart of conventional lane detection systems that obtain lane information from a CMOS image sensor (CIS). In this case, visual information should be obtained to generate a high-resolution image from the CIS to achieve a high accuracy. In the image signal processing unit, the input image is filtered through, for example, smoothing and edge detection [5]. The lanes are detected based on the processed image using the Hough transform. This technique limits the image processing speed and requires many memory blocks [6,7]. Therefore, the conventional lane detection method results in a high-power consumption when used in high-speed cars with high processing speeds [8,9]. Figure 1b shows a system flow chart of the proposed edge detection system. In contrast to the already existing method, shown in Figure 1a, the edge mask can be implemented inside the CIS. Thus, low-power edge detection is realized by simply implementing the existing process, which must perform complex calculations with high-resolution image data in the readout circuits in CIS. Implementing the existing edge mask process into the CIS reduces the high-power consumption caused by processing in the image signal processor. The proposed edge detection process that is performed in the CIS minimizes power consumption while maintaining a high frame rate. We show that compared with conventional edge detection algorithms, such as the Sobel and Prewitt algorithms [10], the proposed edge mask is simpler and shows a reasonable edge detection accuracy. The accuracy of the proposed CIS is higher than that of the CIS with a built-in edge mask, which is presented in [11]. The proposed on-CIS edge detection processing circuits were fabricated in a 110 nm CIS process. The system provides conventional 8-bit images and 7-bit edge detection images. The image resolution is 160 (H) × 120 (V) with 12.8 μm × 12.8 μm of pixel pitch. The experimental results show a power consumption of 9.7 mW, with frame rates of 145 in the CIS mode and 113 in the edge detection mode.

## 2. Proposed CIS Structure

Figure 2 shows a block diagram of the proposed on-CIS edge detection computing system, which consists of pixel array, column-parallel readout circuits, including a row buffer layer (RBL), an edge mask layer (EML) block with an 8-bit single-slope analog-to-digital converter (SS-ADC), a column driver, an 8-bit counter, and a row driver. The RBL stores pixel information and selectively outputs the necessary pixel data in a row. The EML contains the proposed edge mask and uses the data output by the RBL. In addition, the RBL performs correlated double sampling (CDS), which is required in conventional CISs to reduce noise from the pixel and readout circuits. The edge mask in the EML is proposed to achieve high edge detection accuracy, and it can be implemented in the conventional readout circuit of the CIS. From the EML, images in the X-direction and Y-direction (G_x_ and G_y_, respectively) can be obtained to construct “G_x_ + G_y_” images, which are converted into digital codes with an 8-bit SS-ADC. In particular, when the lanes of the road are diagonal with respect to the camera in the vehicle, the proposed edge detection mask provides good G_x_ + G_y_ images, which result in efficient lane detection.

### 2.1. Conventional Edge Detection Mask Algorithm

Figure 3 shows the principle of the mask operation. If the size of the mask is 3 × 3, and the size of the image to which the mask is applied is 3 × 3, the values at the same position are multiplied based on the center pixel (x,y). Subsequently, all the values are summed to obtain the new value, which represents the center pixel M(x,y). The equation representing the operation of the mask is as follows:M(x,y)={A×(x−1,y+1)}+{B×(x,y+1)}+{C×(x+1,y+1)}+{D×(x−1,y)}+{E×(x,y)}+{F×(x+1,y)}+{G×(x−1,y−1)}+{H×(x,y−1)}+{I×(x+1,y−1)}.

If a threshold value is applied to M(x,y), the output of M(x,y) is “high” only when it exceeds a certain value. If the threshold is 0, all codes are “high.” If the threshold is 0.5, the output of M(x,y) is “high” only when it exceeds 127 codes (from 0 to 255 codes). Depending on the mask size, an additional row buffer for storing pixel data may be required.

### 2.2. Proposed Edge Detection Algorithm

The Sobel mask, shown in Figure 4a, is the most commonly used lane detection algorithm. This mask prevents the calculation of false edges in the presence of noise and produces less noise than other masks. However, implementing a 3 × 3 mask in an analog CIS circuit is challenging because of different weights in a column/row. The Prewitt mask, shown in Figure 4b, is simpler than Sobel, because it does not require multiplication by using only 1 or −1 as a weight. However, its implementation in the conventional CIS is also difficult because this also requires three computations for the weights. The Roberts mask, shown in Figure 4c, has been proposed to overcome this difficulty. Due to its size (2 × 2), it is relatively simple, compared to the Sobel and Prewitt masks. In particular, it is suitable for lanes with diagonal lines, because it compares the pixels located diagonally with the center pixel by weighting the former pixels. However, the simpler the mask is, the less accurate its results are. Figure 4d presents the edge detection mask proposed in [11]. Unlike those of other masks, the circuit is simplified by simply comparing columns (X-direction); however, if relatively few data are added, only adjacent pixels are compared, and the noise is high. In the proposed mask, shown in Figure 4e, diagonal information is compared, for example, with the Roberts mask. Furthermore, a wider range of pixels is used when a 3 × 3 mask is applied to calculate the center pixel. The proposed mask reduces noise and data omission, thereby resulting in a higher accuracy.

Figure 5 presents the results of applying the five different types of edge masks shown in Figure 4 to an original image. Pratt’s figure of merit (PFOM) [12,13] was used to analyze the accuracy of the edge detected images. The images for which the edge has been detected are compared with the ideal image to evaluate how many pixels have different values. Therefore, it can be said that the closer the PFOM (%) is to 100, the same as the ideal data. As the Sobel mask is most suitable for diagonal detection [14], Pratt’s figure of merit (PFOM) was used to compare the performance of the five different masks shown in Table 1. After the Sobel mask, the Prewitt mask has the highest PFOM. It performs three calculations based on the center pixel. The PFOM of the proposed circuit is reduced by 0.86, compared to that of the Prewitt mask. Thus, it achieves the second-highest value. While the proposed mask does not achieve the highest PFOM, its performance it similar to that of the Sobel mask; however, it requires only one operation based on the center pixel.

Figure 6 shows that the size of the proposed mask increased from 2 × 2 to 5 × 5. This allows us to determine whether PFOM increases proportionally with the mask size. Table 2 shows PFOM according to the different mask sizes. As the mask size increases from 2 × 2 to 3 × 3, PFOM increases as well. However, as the mask size increases beyond 3 ×3, PFOM is reduced. This result shows that the proposed 3 × 3 is the optimal mask size in terms of PFOM.

### 2.3. Operation of the Proposed CIS with the Built-In Mask

Figure 7 shows the proposed circuits of the RBL and EML. To implement a 3 × 3 mask, the RBL outputs the pixel data (in analog voltage) for the (N − 1)^th^ row and (N + 1)^th^ row based on the N^th^ row of the center pixel. The pixel data stored in the N^th^ row are used twice when processing the center pixels of the (N − 1)^th^ and (N + 1)^th^ rows. Thus, they are used four times. As shown in Figure 7a, the proposed RBL stably stores the pixel data in the capacitor using an operational transconductance amplifier [15]. The pixel data for the three rows are stored in the capacitors connected to nodes 1, 2, and 3, and the final output voltage V_out_ is V_ref_ + △PIX. The values stored in the RBL apply the proposed mask through the EML.

As shown in Figure 7b, the switches G_1_ and G_2_ in the EML are turned on in sequence. G_x_ is implemented by receiving the output of the (M − 1)^th^ column first based on the M^th^ column (which is the center pixel) and by sequentially receiving the output of the (M + 1)^th^ column. By contrast, G_y_ receives the output of the (M + 1)^th^ column and continues to receive the output of the (M − 1)^th^ column. The rows outputted through G_1_ and G_2_ are the (N − 1)^th^ and (N + 1)^th^ rows, respectively. The sequential row outputs are transferred through G_1_ and G_2_ for the proposed edge detection operation (in Figure 4e) according to the following equations:$$G_{x}=\left[\left\{V_{\mathrm{ref}}+\Delta {\mathrm{PIX}}_{\left(N-1,M-1\right)}\right\}-\left\{V_{\mathrm{ref}}+\Delta {\mathrm{PIX}}_{\left(N+1,M+1\right)}\right\}\right]=\Delta {\mathrm{PIX}}_{\left(N-1,M-1\right)}-\Delta {\mathrm{PIX}}_{\left(N+1,M+1\right)},\;G_{y}=\left[\left\{V_{\mathrm{ref}}+\Delta {\mathrm{PIX}}_{\left(N-1,M+1\right)}\right\}-\left\{V_{\mathrm{ref}}+\Delta {\mathrm{PIX}}_{\left(N+1,M-1\right)}\right\}\right]=\Delta {\mathrm{PIX}}_{\left(N-1,M+1\right)}-\Delta {\mathrm{PIX}}_{\left(N+1,M-1\right)}.$$
where ΔPIX is determined by V_reset_ − V_signal_ from a pixel for the CDS operation. Figure 7b compares the pixel values inputted for CDS with the “Ramp” signal. The “Ramp” signal is maintained at V_ref_. When G_x_ or G_y_ is applied, the “Ramp” has a slope with a magnitude in the range of Vmax = V_ref_ + ∆V to V_min_ = V_ref_ − ∆V. Through this process, a positive or negative value based on the center value V_ref_ is outputted, which indicates the direction of the slope between the center pixel and surrounding pixels.

Figure 8 shows the timing diagrams of the conventional CIS and edge detection operation system. During the conventional CIS operation (Figure 8a), the digital code output linearly increases from 0 to 255. However, during edge detection (Figure 8b), the EML receives two-row datasets as input values, and the Ramp signal is +ΔV and −ΔV to detect edges in both directions. Accordingly, the digital output code shows a pattern in which the LSB to MSB-1st code increases from 0 to 127 based on the point at which the MSB code indicates that the phase is converted from low to high.

## 3. Experimental Results

### 3.1. Chip Photograph and Measurement Environment

Figure 9a shows a microphotograph of the chip. The proposed edge detection CIS was fabricated through a 1poly–4Metal 110 nm CIS process. The supply voltages were 3.3, 3.3, and 1.5 V for analog, pixel, and digital circuit blocks, respectively, and the chip area was 5.9 mm × 5.24 mm. The measurement results show that the power consumption of the proposed circuit is 9.4 mW and that the processing speed is 145 fps for the conventional CIS operation. Figure 9b shows the measurement environment. The FPGA board XEM3050 (Xilinx Spartan-3 FPGA Integration Module) was used to check the control signal application and image output to connect the computer. Using the Opal Kelly board from Xilinx, the FPGA was driven by a USB interface, and the successful operation of the circuit was confirmed by checking the final image displayed in the program, “Image Viewer”.

### 3.2. Measurement Results

When an original image (Figure 10a) is captured by the proposed edge detection sensor through the CIS operation, an image, such as that shown in Figure 10b, can be obtained with 145 fps. With edge detection, G_x_, G_y_, and G_x_ + G_y_ images can be obtained, as shown in Figure 10c–e, respectively.

As shown in Figure 11, edge images and top8,9 images of the Hough transform result from (1) Sobel, (2) Prewitt, (3) Roberts, (4) Column comparing, and (5) the Proposed mask are obtained using MATLAB on images from conventional CIS. On the other hand, (6) proposed edge detection images are obtained directly from the proposed edge detection CIS chip. The edge data were output by applying the global threshold (Th = 0.5). For the Sobel and Prewitt masks, three operations were performed to implement the masks, and the edge data in the images are clear. By contrast, for the Roberts and column-comparing masks (with 2 × 2 sizes), only one operation was performed to implement the masks, and the edge data are less sharp, because only the data of adjacent pixels were considered.

By applying the Hough transform to the edge images (Figure 11a), an image with straight lines (Figure 11b,c) is obtained. Table 3 summarizes the degree of recognition of a straight line of each mask. The other masks show errors in terms of the line or noise. Based on the Top 9 (=Top 9 lines recognized as lines), all masks except the column-comparing mask show the same results as the Sobel mask. Therefore, we expect to obtain similar results to those of the Sobel masks when the proposed circuit is used for lane recognition. In addition, we observed that even though the proposed edge detection circuit is implemented inside the low-power CIS, it has similar results as the edge detection of the Sobel mask conducted by MATLAB.

Table 4 shows the PFOMs before the measurement using MATLAB (Pre) and after the measurement from the chip (Post). In the case of measurement from CIS (=Post), since noise exists in the image, PFOM decreases compared to Pre. Each mask has a different sensitivity to noise. In general, a (2 × 2) mask that compares adjacent pixels, that is, a Roberts or Column-comparing mask, is vulnerable to noise because it compares only adjacent pixels and has a large Δ. On the other hand, in the case of a 3×3 mask, as the size of the mask increases, the range of pixels to be reflected increases, so it is relatively robust against noise, resulting in a small Δ. Therefore, the proposed mask is not only resistant to noise but also has the advantage of being able to operate with low power consumption by integrating a simple mask circuit in the CIS.

Table 5 summarizes the performance characteristics of the CISs, including the proposed circuit. The proposed circuit was prepared through a 1poly–4metal 110 nm CMOS process and its chip area is 5.9 mm × 5.24 mm. When processing an image of one frame, the circuit consumes 9.4 mW of power, and it has an operating speed of 113 fps when performing lane recognition and 145 fps when performing the general CIS operation. Table 6 compares the performance characteristics of the proposed and other edge detection masks. The circuit proposed in [8] implements a commonly used mask in the digital domain. To implement the mask, several rows are simultaneously read, and the image edges are screened for vertical, horizontal, and diagonal lines. In [9], the analog signal is directly converted to the frequency domain signal when the built-in mask technique is applied to reduce power consumption and achieve high-speed conversion. The resulting low resolution prevents the analog signal from being used for high-resolution CIS applications.

The mask proposed in [11] performs the conventional CIS operation. Subsequently, the image edge in the vertical direction is detected using XOR and flip-flop operations in the digital domain. The mask is simple to operate and can be used with any ADC; however, it creates noise, because only adjacent pixels are considered. In this study, an edge detection mask was implemented in the analog domain. The proposed mask only detects diagonal lines. Its maximal fps rate is approximately four times that of other circuits. Thus, it is suitable for lane recognition at high driving speeds. In addition, according to the fps rate and supply voltage, the proposed circuit consumes less power. As the MSB code represents a gradient, it can be used for operations that require a phase. Unlike the other circuits, the proposed circuit can be applied in various situations.

## 4. Conclusions

This paper presents a low-power CIS that performs edge detection in the analog domain. By implementing the mask operation process into the CIS, the error due to quantization processing of ADC can be reduced and the PFOM can be reduced to the minimum after the measurement. The Sobel mask, which is the most suitable mask for diagonal detection among the conventional masks, derives its value through three operations. Conversely, the proposed mask requires only one operation and can detect edge data, and its results are similar to those of the Sobel mask (97.24%). Therefore, the proposed CIS can reduce power consumption and accelerate data processing by reducing the processing time. In addition, because it has a driving speed of 113 fps for edge detection (which corresponds to real-time operation conditions), the mask can reduce the risk of vehicles injuring people. As the circuit proposed in this paper can obtain data for lane recognition, the resulting lane recognition sensor with a low-power consumption based on the miniaturization of the chip size is suitable for actual vehicles.

## Figures and Tables

**Figure 1 sensors-21-03713-f001:**
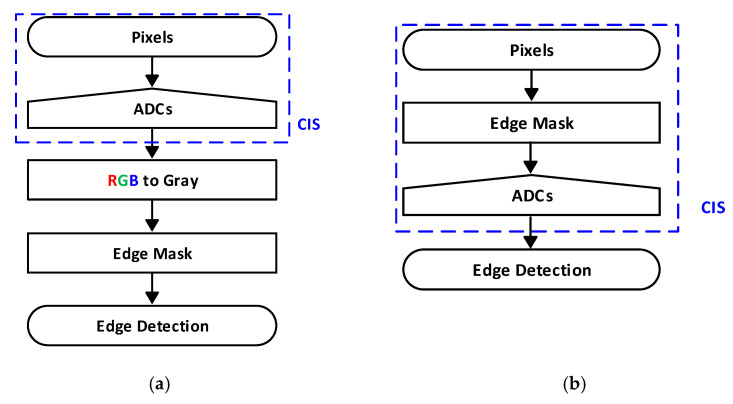
System flow charts: (**a**) conventional edge detection; and (**b**) the proposed edge detection system.

**Figure 2 sensors-21-03713-f002:**
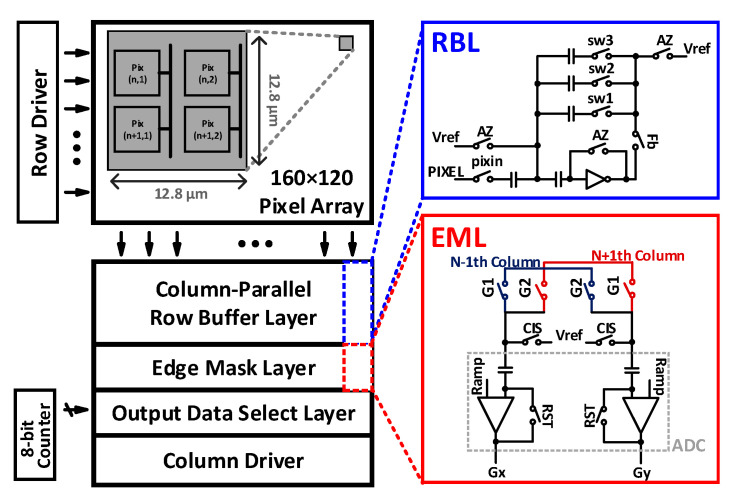
Block diagram of the proposed CIS with the built-in edge-detection mask.

**Figure 3 sensors-21-03713-f003:**
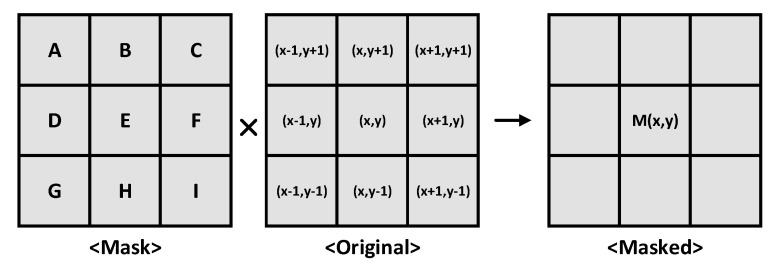
Principle of the mask technique.

**Figure 4 sensors-21-03713-f004:**
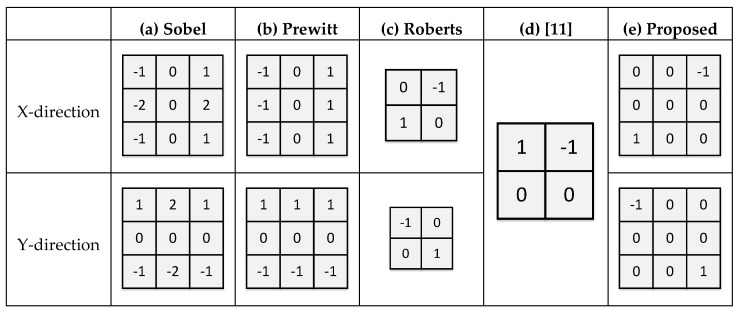
(**a**) The Sobel mask, (**b**) the Prewitt mask, (**c**) the Roberts mask, (**d**) the column-comparing mask [11], and (**e**) the mask built into column circuits.

**Figure 5 sensors-21-03713-f005:**
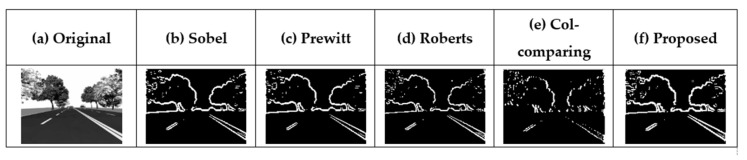
(**a**) Original image and images obtained using the existing mask algorithms: (**b**) the Sobel mask, (**c**) the Prewitt mask, (**d**) the Roberts mask, (**e**) the column-comparing mask [11], and (**f**) the mask built into column circuits.

**Figure 6 sensors-21-03713-f006:**
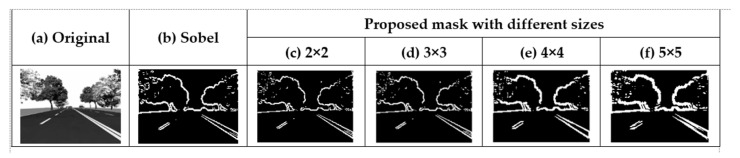
(**a**) Original image and images obtained using the existing mask algorithms: (**b**) Sobel, (**c**) 2 × 2 (Roberts, Proposed), (**d**) 3 × 3 (Proposed), (**e**) 4 × 4, and (**f**) 5 × 5.

**Figure 7 sensors-21-03713-f007:**
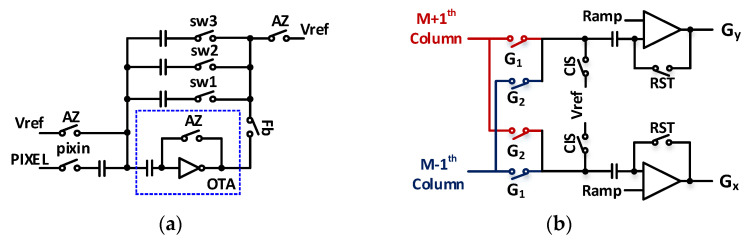
Schematics of (**a**) RBL and (**b**) EML.

**Figure 8 sensors-21-03713-f008:**
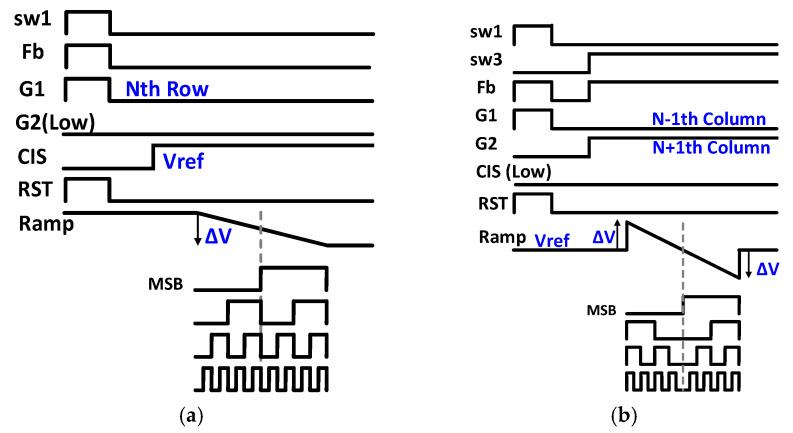
Timing diagrams of (**a**) the conventional CIS operation and (**b**) edge detection operation.

**Figure 9 sensors-21-03713-f009:**
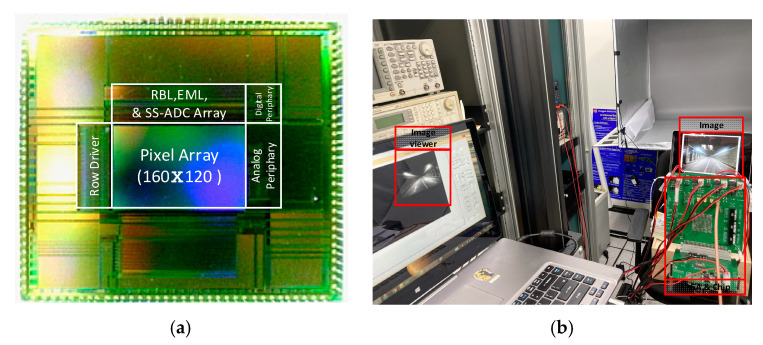
(**a**) Chip microphotograph and (**b**) measurement environment.

**Figure 10 sensors-21-03713-f010:**
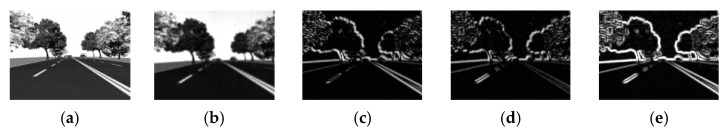
(**a**) Original image and (**b**) CIS image. Edge detection images of (**c**) G_x_, (**d**) G_y_, and (**e**) G_x_ + G_y_.

**Figure 11 sensors-21-03713-f011:**
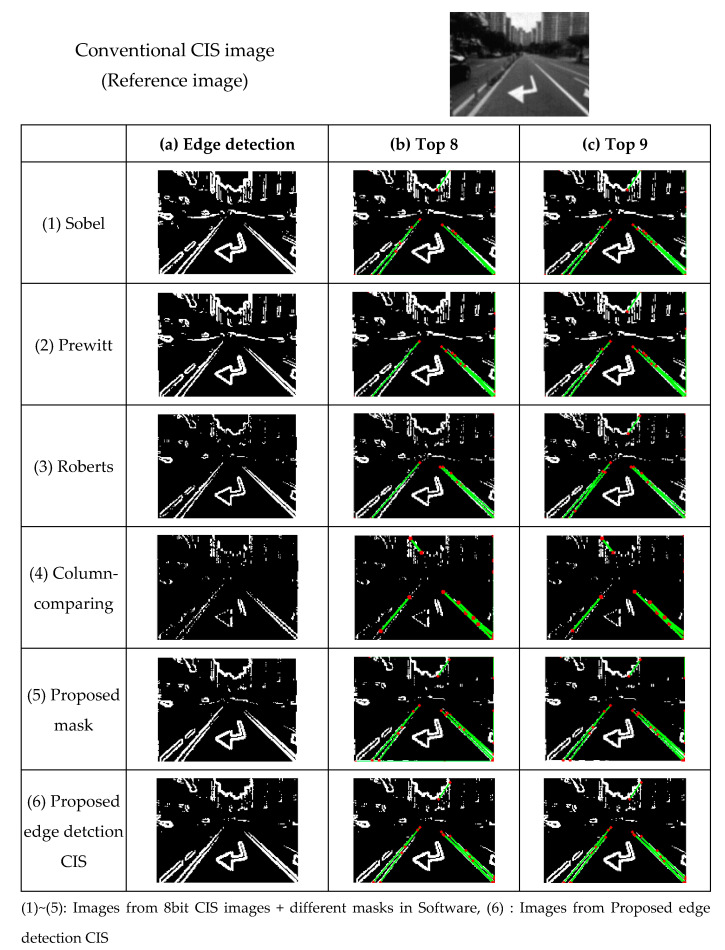
(**a**) Edge detection image and (**b**) Top 8 and (**c**) Top 9 of the lane detection image obtained from the conventional CIS image.

**Table 1 sensors-21-03713-t001:** Pratt’s figure of merit (PFOM) of the different masks.

PFOM (%)	Sobel	Prewitt	Roberts	[11]	Proposed (3 × 3)
Sobel (ref)	100	99.75	97.44	95.38	98.89

**Table 2 sensors-21-03713-t002:** Pratt’s figure of merit (PFOM) at the global threshold value.

PFOM (%)	(c)	(d)	(e)	(f)
Sobel (Ref)	97.44	98.89	96.32	92.54

**Table 3 sensors-21-03713-t003:** Degree of lane recognition of an image that ended the lane recognition (= how many of four center lanes are recognized).

	(1) Sobel		(2) Prewitt		(3) Roberts		(4) Column-Comparing		(5) Proposed Mask		(6) Proposed Edge Detection CIS	
	Top 8	Top 9	Top 8	Top 9	Top 8	Top 9	Top 8	Top 9	Top 8	Top 9	Top 8	Top 9
Line	4	4	3	4	3	4	3	3	4	4	4	4
Noise	1	1	0	1	0	1	1	1	1	1	1	1

**Table 4 sensors-21-03713-t004:** Pratt’s figure of merit (PFOM) of results from five different edge masks.

PFOM (%)	Sobel (Ref)	Prewitt	Roberts	[11]	Proposed Mask
Pre	100	99.75	97.44	95.38	98.89
Post	100	98	94.95	90.51	97.24
Δ	-	1.75	2.49	4.87	1.65

**Table 5 sensors-21-03713-t005:** Performance characteristics of the proposed edge detection CIS.

Technology	110 nm 1P4M CIS Process
Pixel array	160 × 120
Pixel size (${\mu m}^{2}$)	12.8 × 12.8
ADC resolution (bit)	8
Power consumption (mW)	9.4
Max. frame rate (fps)	145 in CIS mode
	113 in edge detection mode
Chip size (${\mu m}^{2}$)	5900 ×5240
Supply voltage (V)	3.3 (analog/pixel)
	1.5 (digital)

**Table 6 sensors-21-03713-t006:** Comparison of performance characteristics of the proposed CIS and other circuits.

	[8]	[9]	[11]	This Work
Edge Image	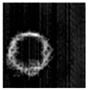	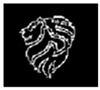	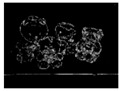	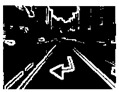
Process	180 nm 1P 5M CMOS	180 nm 1P 4M CIS	90 nm 1P 5M CIS	110 nm 1P 4M CIS
Resolution	70 × 68	105 × 92	1920 × 1440	160 × 120
Pixel pitch (μm)	25.7	8	1.4	12.8
Supply voltage (V)	1.8	1.6	3.3	3.3
Frame/s	28	30	60	113 (edge)145 (CIS)
Power consumption	110 mW	8 mW	9.4 mW (60 fps)	9.4 mW (145 fps)

## Data Availability

The datasets generated from the current study are available from the corresponding author on reasonable request.

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
