# Peer review of "On-CMOS Image Sensor Processing for Lane Detection"

_sensors, 2021, doi:10.3390/s21113713_

Round 1

Reviewer 1 Report

The paper proposed an on-chip CIS edge detection mask for lane detection. The system achieves comparable performance to the conventional method with relatively low power consumption, supported by measurement results. The implementation is interesting, however, further clarification is necessary. I have several comments as follow:

1) In section 2.2, the authors mentioned implementing a 3x3 mask in an analog CIS circuit is challenging. But the proposed mask is also implemented using a 3x3 mask, but with no mention of how it is challenging. Please clarify this.

2) Please elaborate on PFOM. What performance criteria does this FOM focus on? 

3) The author mentioned PFOM is used due to the Sobel mask being the most suitable for diagonal detection. Why is the choice of FOM dependent on which method is most suitable for diagonal detection?

4) I would suggest providing some basic specifications of the conventional CIS used, such as resolution, etc. 

5) In Figure 11, edge detection images are shown. The result would be better if the original image taken by the conventional CIS is included as a reference. 

6) The measurement results using the edge detection sensor shows only the edge detection, without "Top 8" and "Top 9" result. It seems like the proposed sensor can also obtain such a result. I think The comparison of the "Top 8" and "Top 9" results using the edge detection sensor and the conventional sensor is important to evaluate the usefulness of the on-chip circuit. 

7) Please clarify how do PFOM pre and post-measurement are obtained. Why is PFOM (post) generally smaller?

8) Please explain why is there a reduced quantization when the edge mask is implemented into the circuit. 

9) The pixel size in Table 5 is 3.2 x 3.2 um, but "pixel pitch" in Table 6 is 12.8 um. Please clarify the difference between these two. 

10) Is the column size of the RBL and EML the same as the pixel size?

11) The proposed sensor can achieve a frame rate of 113fps but on a relatively small resolution of 160 x 120. Could you provide references to the commercially available sensors used for lane detection and their resolution, or is there any standard for sensor resolution for LDWS? How much the frame rate will be if the sensor is of QVGA or VGA? 

Minor comments:
1) Line 13: The image resolution is shown as 160 x 120. Please specify which number is vertical and horizontal.

2) Line 41: I would suggest adding references for Sobel and Prewitt algorithms. 

3) Fonts in the figures are too small, especially in Figure 2 and Figure 7. I would suggest making the fonts more readable. 

4) Line 124: Typo? RBM -> RBL

5) Is sw1 and sw3 in Figure 8 the same as s1 and s3 in Figure 7? If they are the same I would suggest using the same naming.

Author Response

First of all, we are really thankful for the efforts to provide us your valuable comments. Your feedbacks are very much appreciated since they improve not only the quality of the paper but also our work. All of your comments are analyzed and replied in this letter. Further, a few parts of the paper are rewritten and improved in the first revised version, according to the editor and reviewers’ comments. We hope that the revised paper will have a good result. Please see the attachment file.

Reviewer 2 Report

  • In Fig. 8 please correct the Chinese character that appears several times instead of the delta symbol in delta V
  • In Fig. 10, the result images from c) and d) should be switched in between to represent Gx and Gy respectively
  • You mentioned that “The edge data were outputted by applying the global threshold”. What is the threshold value that was used and how was it chosen?
  • The method of selecting Top 8 or Top 9 of the Hough lines for lane recognition is not robust at all. What happens, for example, when there are many poles (pillars) on both left and right side of the road?
  • The comparison of the images from Table 6 is irrelevant. You should have compared (if possible) the edge detection results from multiple sensors when using the same input image.
  • Somehow, many figures are misaligned with the rest of the text from the manuscript, please fix them.

Author Response

(The authors gave the same response as above.)

Round 2

Reviewer 1 Report

The authors have provided sufficient modifications and clarifications. I have no further comments. 

Reviewer 2 Report

-